# Aflatoxin B_1_ Metabolism of Reared *Alphitobius diaperinus* in Different Life-Stages

**DOI:** 10.3390/insects13040357

**Published:** 2022-04-06

**Authors:** Nathan Meijer, Rosalie Nijssen, Marlou Bosch, Ed Boers, H. J. van der Fels-Klerx

**Affiliations:** 1Wageningen Food Safety Research (WFSR), P.O. Box 230, 6700 AE Wageningen, The Netherlands; rosalie.nijssen@wur.nl (R.N.); ed.boers@wur.nl (E.B.); ine.vanderfels@wur.nl (H.J.v.d.F.-K.); 2Ynsect NL Nutrition & Health B.V., Harderwijkerweg 141B, 3852 AB Ermelo, The Netherlands; marlou.bosch@ynsect.com

**Keywords:** aflatoxins, mycotoxins, edible, insects

## Abstract

**Simple Summary:**

The contamination of food and feed by aflatoxins causes major health and economic consequences, especially in sub-Saharan Africa. Aflatoxins can cause liver cancer and other health issues. Certain reared insect species that are being considered as an alternative protein source have been found to metabolically convert aflatoxins in their diet. The aim of this study was to investigate the metabolism of aflatoxin B_1_ by Lesser Mealworm to determine if this chemical is present in the substrate on which this species is reared. Possible differences in metabolic rate and/or metabolic compounds in three different life-stages of this species were investigated. We observed no negative effects on growth and survival, suggesting that the Lesser Mealworm is very resistant to aflatoxin B_1_. Analyses of the larvae and the excreta after the experiment showed that aflatoxin B_1_ and known aflatoxin derivatives (metabolites) were not present in the insects, and concentrations in the excreta were up to 40% of the spiked concentrations in the feed. It was not clear whether the other proportion that could not be quantified was completely detoxified or converted into unidentified (toxic) metabolites. If detoxification can be verified, then the Lesser Mealworm is a very suitable candidate for the rearing on aflatoxin-contaminated feed materials.

**Abstract:**

The presence of carcinogenic aflatoxins in food and feed is a major issue. In prior studies, aflatoxin B_1_ (AfB_1_) and known primary metabolites were absent from Lesser Mealworm (LMW, *Alphitobius diaperinus*) reared on contaminated diets. LMW is a promising alternative protein source. The objectives of this stu\dy were to determine whether LMW can be reared on AfB_1_-contaminated feed in each life-stage, and to gather more insight into potential metabolites formed. Results suggested no adverse effects in terms of survival/growth when three stages of LMW (larvae, pre-pupae, beetles) were exposed to feed containing AfB_1_ concentrations of 200 and 600 µg/kg for 48 h. Insect and frass samples were analyzed by LC-MS/MS and high-resolution MS to, respectively, quantify concentrations of AfB_1_ and its major metabolites, and determine secondary metabolites. No AfB_1_ or major metabolites were quantified in the insect samples. Mass balance calculations showed that up to 40% of spiked AfB_1_ could be recovered in the frass, in the form of AfB_1_, aflatoxicol and AfM_1_. HRMS results suggested the presence of additional metabolites in the frass, but, due to lack of commercially available reference standards for these compounds, exact identification and quantification was not possible. More research is needed to verify the absence of toxicity.

## 1. Introduction

Aflatoxin B_1_ (AfB_1_) is a highly toxic contaminant produced by certain fungi that grow on crops and stored food and feed materials, such as peanuts, rice and corn [1,2]. AfB_1_ is carcinogenic [3,4], and it has been correlated to a number of indirect effects, such as stunting in children [5,6]. It can also cause direct adverse health effects in livestock [7,8,9] and humans [5,10] at high consumed concentrations. Exposure to AfB_1_ is a major concern, especially in Sub-Saharan Africa, in terms of disease burden and economic effects [1,11,12,13]. The parent compound AfB_1_ is metabolized in a number of ways, most notably into aflatoxin M_1_ (AfM_1_) by cattle [14,15]. Although the toxicity of these metabolites is lower relative to the parent compound AfB_1_, their presence in food and feed is still a reason for concern [16].

Certain insect species that have received interest for their potential as alternative protein sources for feed and food seem to metabolize AfB_1_. Research has focused on black soldier fly larvae (*Hermetia illucens* (L.), Diptera: Stratiomyidae) [17,18,19,20] and, to some extent, yellow mealworm (*Tenebrio molitor* (L.), Coleoptera: Tenebrionidae) [17,21] and Lesser Mealworm (LMW, *Alphitobius diaperinus* (Panzer); Coleoptera: Tenebrionidae) [20].

LMW, in particular, is an interesting candidate for the valorization of AfB_1_-contaminated crops, because LMW grow well on grain-based diets [22]. In the study by Camenzuli et al., approximately 56–80% of spiked AfB_1_ could be recovered from the residual material post-trial, when the LMW were exposed to 0.008 to 0.43 mg/kg in the diet. The presence of the metabolites aflatoxicol (AfOL), aflatoxin P_1_ (AfP_1_), aflatoxin Q_1_ (AfQ_1_) and AfM_1_ was also analyzed—but only AfOL (0.3%) and AfM_1_ (2.5%) could be quantified. As such, it was concluded that a proportion of the spiked AfB_1_ must have undergone different or additional metabolic steps [20].

The objective of this study was to determine the metabolism of AfB_1_ by LMW if this contaminant is present in the feed on which this species is reared. We hypothesized that there may be a difference in metabolic rate and/or products, depending on the life-stage of this species. Three life-stages of *Alphitobius diaperinus*, being larvae, pre-pupae and beetles, were exposed to spiked AfB_1_ in the diet. Insect and frass samples were analyzed by liquid chromatography–mass spectrometry (LC-MS/MS) as well as liquid chromatography–high resolution mass spectrometry (LC-HRMS) to gather more insight on the metabolites formed.

## 2. Materials and Methods

### 2.1. Substrate Preparation

Three 400 g batches of LMW feed were used for this experiment. The feed was spiked with AfB_1_, using 100 μg/mL laboratory standard (Romer Labs, Getzersdorf, Austria), at intended concentrations of 200, 600 and 0 (control) μg/kg. After spiking the feed with the AfB_1_, a slurry was created by adding 800 mL of methanol in parts to each batch in order to achieve a homogenous distribution of the AfB_1_. The same procedure to create a slurry was also followed for the control treatment. After mixing the slurry manually, it was placed in an aluminum container and left overnight in a fume hood for the methanol to evaporate. The next day, the remaining feed was weighed to determine moisture loss and, subsequently, grinded (Retsch Grindomix GM 200) and mixed again (Stephan UMC 5 electronic). Finally, the feed was distributed in separate containers so that each container contained 100 g of spiked, dry feed to be added to the replicate containers later. From each batch, 1 g (*n* = 10) and 2.5 g (*n* = 1) aliquots were taken for later analyses of homogeneity and concentration of AfB1 in the spiked feed, respectively.

### 2.2. Experimental Procedures

*Alphitobius diaperinus* larvae were reared on the AfB_1_-spiked feed till three different life-stages: larvae, pre-pupae and beetles. Each treatment was performed in triplicate. All insects were reared from the same batch of eggs to avoid any biological variation between the tested life-stages. After 13 days, for each of the replicates, 30 g of the larvae were cleaned and moved into the replicate container. Per container, 25 g of the respective spiked/un-spiked feed was fed in two equal portions (once per 24 h). The dimensions of the replicate container were 19 cm × 11.5 cm; 950 cc (Dampack International BV, Werkendam, The Netherlands). This process was repeated at 26 days for 30 g of the pre-pupae, and at 46 days for 30 g of the beetles. After 48 h of exposure to the spiked substrate, the insects of each replicate were cleaned and moved to a separate replicate container containing 25 g of un-spiked substrate. The frass remaining in each replicate container was emptied into a closed container and frozen for transport and subsequent analysis. This step was performed so that the insects would empty their gut contents, to avoid contaminated substrate in the guts contributing to the total AfB_1_ load. Finally, after 24 h on the un-spiked substrate, the insects of each treatment were thoroughly cleaned, placed into a plastic container (dimensions 11.5 × 11.5 × 6 cm) and frozen at –18 °C for subsequent analysis. Cleaning was done by shaking the insects over a 1 mm sieve, so that all attached fecal and feed particles were removed from the animals. The freezing step killed the insects. The weight of the insects plus residual material post-exposure was measured. The weight of the residual material was determined by subtracting the weight of the larvae and container (determined prior to the experiment) from the total weight of the replicate container (i.e., still containing the larvae) directly after exposure.

### 2.3. Chemical Analyses

#### 2.3.1. Extraction

Extraction followed the procedure described by Camenzuli et al., 2018 [20], in accordance with the QuEChERS method. In short, 2.5 g of sample material was weighed, to which 7.5 mL water, 10 mL of extraction solvent (acetonitrile (Biosolve HPLC Supra Gradient)/acetic acid (Merck, Darmstadt, Germany) 99:1 (*v*/*v*)) and 25 μL of a 10µg/mL solution of ^13^C-caffeine (Sigma Aldrich, St. Louis, MO, USA) was added. After shaking (Heidolph, Reax 2), 4 mg of dried magnesium sulfate (MgSO_4_) was added, and the mixture was vortexed for 1 min and subsequently centrifuged (10 min at 3500 rpm, Thermo Scientific SL 40 R, Thermo Fisher Scientific, Waltham, MA, USA). Finally, 200 μL of sample extract was transferred to polypropylene vials, and 200 μL water was added. Samples were stored at 4 °C until analysis. Quantification took place through standard addition.

#### 2.3.2. LC-MS/MS Analyses

The LC-MS/MS system consisted of a Waters Acquity injection and pump system (Waters, Milford, MA, USA) and an AB Sciex QTRAP 6500 triple quad system equipped with an electrospray ionization (ESI) source, which was operated in positive mode. Instrumental MS/MS parameters are shown in the Appendix A. LC separation was performed by an Acquity UPLC HSS T3 1.8 μm 100 × 2.1 mm column (Waters, Milford, MA, USA). Remaining LC/MS-MS procedures were identical to Camenzuli et al., (2018).

Eluent A consisted of H2O and eluent B was composed of MeOH:H2O 95/5 (*V*/*V*). Both eluents contained 1 mM ammonium formate and 1% formic acid. The LC eluent gradient started with an initial period of 1 min at 100% A; the proportion of B was linearly increased to 50% at 2 min and followed by a linear gradient of 100% B at 8 min and was kept for 2 min. At 10.5 min, the initial conditions were restored and the method ended at 12.5 min. The flow rate was 0.4 mL/min, the column temperature was 40 °C, and the injection volume 5 μL. The conditions set for electrospray ionization were as follows: spray voltage 4.0 kV, temperature, 400 °C; Ion Source Gas 1 and 2 were both set at 50 arbitrary units.

Analyst software v1.7.1 (Sciex, Framingham, MA, USA) and MultiQuant v3.0.2 (Sciex, Framingham, MA, USA) were used to analyze the LC-MS/MS data obtained. All analyzed concentrations were corrected for recovery.

#### 2.3.3. Quality Control

A matrix-matched calibration was prepared in a blank extract of each of the matrices (substrate, larvae and residual material) to calculate the AfB1 (and metabolite) concentrations in the samples. Recovery percentages were used to adjust calculated results.

Determination of the homogeneity of spiked substances in the diet was done in accordance with ISO standard 13528:2015. Specifically, of both treatments (200 and 600 μg/kg AfB1), the *n* = 10 aliquots mentioned in the section on substrate preparation above were analyzed in duplicate by LC-MS/MS (see paragraph above). Outliers were calculated and removed, and a relative standard deviation (RSD) of 25% was considered acceptable.

#### 2.3.4. LC-HRMS Screening

The extracts of larvae and residue material as prepared for LC-MS/MS measurement of known AfB1 metabolites were also used for the screening of unknown AfB_1_ metabolites. The measurements were performed using LC-Q-Orbitrap-MS, using the method previously described for pesticide analysis [23]. Additionally, one frass material sample and one insect sample of the 600 µg/kg treatment level for each of the three tested life-stages was analyzed using the same chromatography, but with full scan analysis combined with top5 DDMS2. DDMS2 results in cleaner MS2 spectra as compared to the vDIA method, but MS2 spectra are only triggered for the top 5 highest signals measured in full scan in each scan cycle, whereas vDIA provides spectra for all signals, but of lower quality.

With the use of LC-Q-Orbitrap-MS technology, an untargeted analysis is used to detect a wide range of compounds and perform retrospective data analysis. Samples were only analyzed in positive ionization mode, since AfB_1_ and all known potential metabolites could be detected in this mode.

Data processing was performed using Compound Discoverer 3.1 (Thermo Scientific, Waltham, MA, USA). The workflow used for processing with the data resulting from measurements is shown in Figure 1. Using this software, the samples were grouped by treatment and insect life-stage. The control samples were used to perform a background subtraction for data reduction. The signal intensity threshold for the detection of features was set to 1 × 10^6^. Insects and frass samples were processed separately.

The Compound Discoverer workflow used consisted of an ‘expected compounds’ search and an ‘unknown’ search. The expected compounds search calculated possible metabolites based on the molecular structure of AfB_1_, including phase 1 and phase 2 metabolism and de-alkylation steps. The unknown search aimed to annotate as many features as possible present in the data files. The mass, MS2 spectrum and retention time were extracted from the raw data and the elemental composition of the compounds was calculated based on mass and isotope pattern. Using the mzCloud and Chemspider databases, tentative identifications of the detected features were assigned. Subsequently, mzLogic (in silico fragmentation prediction) was used to score the match between the spectrum and in silico spectrum of a molecule.

To evaluate the detected features, several filters were used. All peaks with intensity less than 5 × the intensity in control samples were considered background. In at least one sample group, the group CV should be ≤20%. These settings strongly reduced the number of features to be manually evaluated.

Molecular identification confidence scoring was done using the method described by Schymanski et al., (2014) [24]. In short, five levels of identification confidence are distinguished. The highest confidence level is 1, ‘confirmed structure’, meaning confirmation using measurement of a reference standard. Level 2 is a probable structure, which is further divided into level 2a (‘library’; unambiguous spectrum–structure match) and level 2b (‘diagnostic’; single likely candidate). Levels 3 (‘Tentative candidate(s)’), 4 (‘Unequivocal molecular formula’) and 5 (‘Exact mass of interest’) are of decreasing confidence.

### 2.4. Statistical and Data Analyses

#### 2.4.1. Statistical Analyses

Differences between the different treatments in terms of the weight of the harvested insects were statistically compared for each of the three tested life-stages of the LMW (larvae, pre-pupae, beetles). This was done using a pairwise Kruskal–Wallis test with a significance level (α) of 0.05. This non-parametric test was chosen because a normal distribution could not be assumed due to the sample size. The same statistical test (Kruskal–Wallis, α = 0.05) used for determining differences in the weight of insects and frass was used to determine significant differences in AfB_1_ concentrations in the frass as related to the exposure concentration (200/600 µg), between the different life-stages. All statistical tests were performed in SPSS Statistics for Microsoft Windows (version 25.0.0.2, IBM Corp., Armonk, NY, USA).

#### 2.4.2. Molar Mass Balance Calculations

Molar mass balance calculations were performed to compare the presence of AfB_1_ and its metabolites pre- and post-experiment.

The weight of the cleaned larvae after 24 h on the control diet was used for mass balance calculations. The weight of the residual material for the mass balance was calculated by subtracting (1) the weight of the empty replicate container and (2) weight of cleaned larvae on day 7, from the weight of the replicate container after 7 days, containing larvae and residual material. The weight of residual material was calculated in this manner to correct for any residual material that may have been lost during sieving of the larvae. The post-experiment AfB_1_ mass (µg) (larvae and residual material) was calculated based on the concentration (µg/kg) and weight of the matrix (g), and compared against the mass of the spiked AfB_1_ pre-experiment. Concentrations of metabolites below the limit of quantification (<LOQ) were presumed to be zero, and therefore not included in mass balance calculations.

Since the molecular weight of AfB_1_ metabolites differed from the parent compound, the molar mass balance calculations were performed by first translating the weight in µg to mol as a unit. This was done by dividing the weight of the metabolite or parent compound in the matrix (in g) by the molecular weight (in g/mol). The molecular weights of AfFB_1_ and analyzed metabolites were: AfB_1_ (312.27), AfOL (314.29), AfM_1_ (328.27), AfP_1_ (298.25) and AfP_1_ (328.27). The total molar mass of post-experiment compounds (larvae + residual material) was expressed as a percentage of the mass of AfB_1_ pre-experiment (substrate).

## 3. Results

### 3.1. Quality Control

The recovery percentages of analyzed compounds using LC-MS/MS are shown in Appendix A.

Results of quality control analyses in the substrate are shown in Appendix A. For both of the spiked treatments, the distribution of AfB_1_ in the substrate was acceptably homogenous. The mean analyzed AfB_1_ concentration in the first treatment (intended: 200 µg/kg) was 187.98 ± 4.00 µg/kg; for the second treatment (600 µg/kg), this was 557.26 ± 7.96 µg/kg.

### 3.2. Larval Yield

The weight of insects and frass on the days that these were collected are shown in Table 1 for each of the exposure concentrations (0, 200, 600 µg). Differences in the weight of the insects between the tested exposure concentrations were not significant, for each of the three life-stages (*p* > 0.05).

### 3.3. Concentrations and Molar Mass Balance

Analyzed concentrations of the parent compound AfB_1_ and major metabolites (AfOL, AfP_1_, AfQ_1_, AfM_1_) in the insects and residual material are shown in Table 2. Neither AfB_1_ nor any of the analyzed metabolites could be quantified in the insects, regardless of life-stage (larvae, pre-pupae, beetles) and exposure concentration (200, 600 µg/kg). The metabolites AfP_1_ and AfQ_1_ could not be quantified in the frass either. The concentrations of AfM_1_ in the residual material were relatively low, but consistently higher in the 600 µg/kg treatment than in the 200 µg/kg treatment. Finally, the concentrations of AfOL were <LOQ in the frass of all 200 µg/kg treatments, and slightly above LOQ in the 600 µg/kg treatments.

The molar mass balance of the LMW experiment is shown graphically in Figure 2. Since AfB_1_ and the analyzed metabolites could not be quantified in the insects, only the contribution to the molar mass balance of these compounds via the frass is shown. The mean total post-experiment mass—as a percentage of the pre-experiment mass—was between 26% (D14, 200 µg) and ~40% (D27, both treatments). The contribution of metabolites AfOL and AfM_1_ to the masses was low. Thus, in all cases, the majority (>50%) of spiked AfB_1_ could not be recovered post-experiment. In addition, these data show some variation in the capacity of different LMW life-stages to metabolize AfB_1_. The relatively highest proportion of post-experiment AfB_1_ appears to have been recovered from the pre-pupal larvae, suggesting that the metabolic rate was lowest in this stage.

### 3.4. Secondary Metabolites

#### 3.4.1. Frass

Using the unknown search in the residue samples with the filters mentioned in the Materials and Methods, 743 features were detected (without filtering 6384). This was a too high number of features to check manually and, therefore, an additional filter was implemented. The added filter was a ratio between the 200 µg/kg and 600 µg/kg treatments. This ratio was set to be between 0.5 (0.33 + 50%) and 0.0167 (0.33 − 50%) in at least one sample group. With this additional filter, 17 features remained, as shown in Table 3. Amongst these detected features was AfB_1_ (molecular weight: 312.277). The other detected peaks were manually evaluated and considered to be not related to AfB_1_.

In the expected compounds search, six m/z–retention time combinations were detected after filtering. For some combinations, several metabolite options were possible—in total, 14 options. Two of the six signals were in the noise area. For the other detected masses, fragmentation did not match the expected fragmentation pattern. After evaluation, four signals remained, shown in Table 4.

The presence of AfB_1_ was confirmed via the reference standard, and the corresponding confidence level was therefore 1. The MS/MS spectrum for this compound is shown in Figure A1. The spectrum of the second metabolite (AfB_1_ + O) is shown in Figure A2. For this metabolite, the position of the oxidation on the molecular structure is not known; therefore, the confidence level is 3. Although its structure is thus unknown, it can be inferred that it is neither AfG_1_ nor AfQ_1_ with retention times of 8.21 and 7.87 min, respectively. As with AfB_1_, the presence of AfM_1_ was confirmed by the reference standard. The spectrum for this metabolite is shown in Figure A3. Finally, the spectrum of the AfB_1_ + H_2_SO_4_ metabolite is shown in Figure A4. This metabolite has confidence level 5; it is a mass of interest, but the molecular formula and structure are not unequivocal.

#### 3.4.2. Insects

For the insect samples, the same approach as for the residue material samples was used. In the unknown search, after filtering, 105 features remained. These were manually checked and considered to be natural compounds (e.g., fatty acids), not features related to AfB_1_ metabolites. In the expected compounds search, nine masses were detected. These were manually checked. Three of these nine masses were visually determined to be noise, and not peaks with corresponding compounds. For the remaining peaks, the fragmentation pattern was evaluated, and these did not match the expected fragmentation for AfB_1_-related compounds. Therefore, it was concluded that, in the insects, no AfB_1_ or metabolites were detected.

## 4. Discussion

To the best of our knowledge, this was the first study to assess the effects of AfB_1_ on multiple life-stages of *Alphitobius diaperinus*. Results showed no significant differences between the larvae, pre-pupae and beetles in terms of survival, growth and concentrations of AfB_1_ and its metabolites in the insects. This suggests that this species is capable of being reared on AfB_1_-contaminated diets in every stage of its lifecycle without any adverse effects on the insects.

Concentrations of AfB_1_ and measured metabolites were below the limit of quantification (LOQ) in all LMW insect samples. The molar mass balance of the LMW experiment showed that approximately 26% to 40% of AfB_1_ could be recovered post-experiment, suggesting that the AfB_1_ that had been spiked in the diet was largely metabolized. In the frass, only the metabolites AfOL and AfM_1_ were present, both at relatively low concentrations. The 60% to 74% of AfB_1_ that could not be recovered can therefore not be explained by primary metabolic products, suggesting that secondary metabolites have been formed. In the results of Camenzuli et al., (2018) [20], approximately 56% and 80% of AfB_1_ could be recovered from an experiment that focused solely on the larval stage of LMW. In this study, recovery was 26% to 40% in the frass of the three assayed life-stages; the slight discrepancy in recovery between these studies may be attributed to analytical uncertainty.

In the HRMS results, AfM_1_ was confirmed as a formed metabolite—but AfOL was not. Instead, the presence of another oxidated metabolite (+O) with a different retention time than AfM_1_, AfG_1_ and AfQ_1_ was determined, as well as a metabolite that underwent hydration and sulfation (+H_2_O_4_S). Since these secondary metabolites could not be quantified, it is unclear to what extent they are responsible for the missing proportion of spiked AfB_1_, nor can any conclusions be drawn on their relative toxicity. The differences in detected metabolites are most likely due to the lower sensitivity of the HRMS method. When using LC-MS/MS, a dedicated method is used, focusing on the known metabolites. All method settings are optimized for these selected compounds, whereas with HRMS, a very generic method is used to be able to measure a wide range of compounds with different physio-chemical properties.

## 5. Conclusions and Recommendations

The results of this study strongly suggest that AfB_1_ and its metabolites are absent from the insects, in line with previous studies. However, the molar mass balance remains incomplete at this time. The HRMS analysis confirmed the presence of AfB_1_, AfM_1_ and one additional AfB_1_ + O metabolite that is not AfG_1_ or AfQ_1_—but the exact nature of this metabolite could not be determined. As such, the possibility that potentially toxic metabolites may have been formed cannot be ruled out. Therefore, we would recommend follow-up studies to directly focus on determining the toxicity of insects that have been exposed to mycotoxins in their diet. This can be done by means of additional in vivo experiments, but new ‘organ on a chip’ and ‘in silico’ technologies may offer an appropriate, more ethical alternative.

## Figures and Tables

**Figure 1 insects-13-00357-f001:**
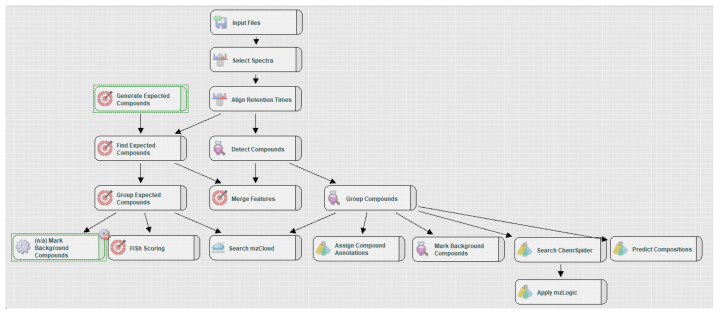
Workflow for processing HRMS data.

**Figure 2 insects-13-00357-f002:**
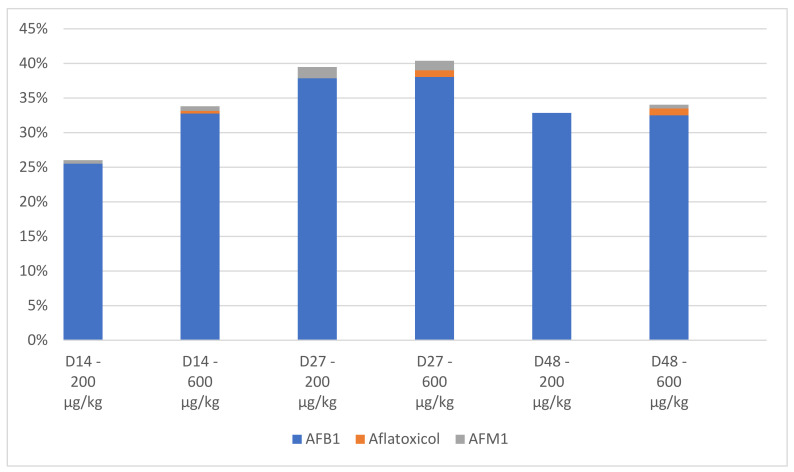
Concentration of aflatoxin B1 (AfB1), aflatoxicol and aflatoxin M1 (AfM1) in the frass, as a percentage of the analyzed concentration in the diet per treatment and collection day.

**Table 1 insects-13-00357-t001:** Weight of frass and insects on experimental day on which the matrix was collected, for each of the three treatments. Mean and standard deviation.

Matrix and Experimental Day	Treatment (Spiked Concentration)		
	0 µg (Control)	200 µg	600 µg
Frass D14	22.33 ± 0.58	21.00 ± 1.00	24.00 ± 1.00
Larvae D15	38.33 ± 0.58	38.33 ± 0.58	32.00 ± 1.00
Frass D27	27.00 ± 1.73	27.33 ± 2.52	27.00 ± 1.73
Pre-pupae D28	32.67 ± 0.58	33.00 ± 0	33.33 ± 0.58
Frass D48	29.67 ± 0.58	27.67 ± 0.58	26.67 ± 1.15
Beetles D49	31.00 ± 0	31.67 ± 0.58	32.0 ± 0

**Table 2 insects-13-00357-t002:** Concentrations of aflatoxin B_1_ (AfB_1_) and metabolites aflatoxin M_1_ (AfM_1_), P_1_ (AfP_1_), Q_1_ (AfQ_1_) and aflatoxicol (AfOL) per matrix and day of collection, and treatment (concentration in feed). Concentrations in µg/kg. Median and standard deviation, or limit of quantification (LOQ) value if all samples below this value.

Matrix and Collection Day	Intended AfB1 in Feed	Analyzed Concentrations (µg/kg)				
		AfB1	AfM1	AfP1	AfQ1	AfOL
Frass D14	0 µg	<2	<2	<4	<8	<4
	200 µg	117.2 ± 0.6	2.2 ± 0.0	<4	<8	<4
	600 µg	389.8 ± 30.6	7.6 ± 2.3	<4	<8	5.6 ± 1.1
Larvae D15	0 µg	<2	<2	<2	<2	<2
	200 µg	<2	<2	<2	<2	<2
	600 µg	<2	<2	<2	<2	<2
Frass D27	0 µg	<2	<2	<4	<8	<4
	200 µg	123.5 ± 5.6	5.3 ± 0.6	<4	<8	<4
	600 µg	371.5 ± 7.8	13.4 ± 0.8	<4	<8	8.8 ± 1.5
Pre-pupae D28	0 µg	<2	<2	<4	<2	<2
	200 µg	<2	<2	<4	<2	<2
	600 µg	<2	<2	<4	<2	<2
Frass D48	0 µg	<2	<2	<4	<8	<4
	200 µg	107.6 ±5.7	<2	<4	<8	<4
	600 µg	327.7 ± 10.5	5.2 ± 0.1	<4	<8	10.2 ± 1.0
Beetles D49	0 µg	<2	<2	<4	<2	<2
	200 µg	<2	<2	<4	<2	<2
	600 µg	<2	<2	<4	<2	<2

**Table 3 insects-13-00357-t003:** Features detected in ‘unknown search’ after filtering. Molecular weight and retention time (RT) (min).

Molecular Weight	RT (min)
199.2303	11.19
212.0687	6.04
213.2096	8.24
215.2251	8.66
227.2253	8.30
271.2514	10.10
273.267	10.45
287.2462	12.01
292.1287	8.52
294.1584	5.74
303.1224	6.47
308.1992	10.04
312.2798	8.83
318.2177	12.12
327.2081	12.44
487.3881	12.92
591.4511	14.00

**Table 4 insects-13-00357-t004:** Features detected in ‘expected compounds search’ after filtering and evaluation.

Conf. Level	Tentative Name	Predicted Formula	Mol. Mass (Da)	Transformations	Composition Change	RT (min)
1	AfB_1_	C_17_H_12_O_6_	312.0634	n/a	n/a	8.83
1	AfM_1_	C_17_H_12_O_7_	328.0583	Oxidation	+(O)	8.03
3	AfB_1_ + O	C_17_H_12_O_7_	328.0583	Oxidation	+(O)	7.04
5	n/a	C_17_H_14_O_10_S	410.0308	Hydration, Sulfation	+(H_2_O_4_S)	8.66

## Data Availability

Data are contained within the article or Appendix A.

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
