# Peer review of "Aflatoxin B1 Metabolism of Reared Alphitobius diaperinus in Different Life-Stages"

_insects, 2022, doi:10.3390/insects13040357_

Round 1
Reviewer 1 Report
Good study
Very few comments inside the article
supplementary tables Table S1 - Captions in all letters Q1,... as footnote to table

Author Response
Dear reviewer,
We thank you for your kind words and review of our manuscript. Below we have responded to your comments indicated in the manuscript.
Kind regards,
On behalf of my co-authors,
Nathan Meijer
Line 90: how many replications?
The number of replicates has been added as requested.
Table S2 must follow table 1 and so on and table 2 before table 1
All tables with S before the number are provided in the supplementary materials. The sequence of tables in supplementary materials is in line with mention in the manuscript, and this sequence is separate from sequence of the tables in the manuscript (without the S). At request of reviewer 2, a table containing results of weight of the insects was moved to the manuscript; the numbering of tables in the manuscript was amended accordingly.
In table 2, for molecular weight: use only 2 decimal places
In table 3, for molecular weight: two decimal places
Reported molecular weights are accurate to 5 ppm, so we believe 4 decimals to be correct. Due to the accuracy of the method to determine molecular weights, we were able to generate molecular formulas based on precise mass figures and isotope patterns, which is the primary advantage of the HRMS method.
supplementary tables Table S1 - Captions in all letters Q1,... as footnote to table
Footnotes have been added to the Table S1 in the revised supplementary materials as requested.
Reviewer 2 Report
The contamination of food and feed by aflatoxins is one of the most common quality defects, consuming aflatoxins may cause sincere health consequences, including nausea, vomiting, growth retardation, and liver injury. Using insects to make use of aflatoxin-contaminated food and feed materials and produce high-quality insect protein may have a positive economic impact.
The manuscript investigated the metabolism of aflatoxin B1 in Lesser Mealworm (LMW) in different life-stages. It helps better understand the tolerance of LMW to aflatoxins. The study is well designed and the manuscript is well prepared. However, there are several minor questions and potential improvements that need to be addressed.
- “ Alphitobius diaperinus” or “Less Mealworm”, both names work but it needs to be consistent throughout the manuscript to improve readability, the name “ Alphitobius diaperinus” should be replaced with “lesser Mealworm”, and Lesser Mealworm as a name of species should be capitalized.
- Experiment procedures, add a summary sentence indicating how many replicates were used in this study at Line 88 before “All insects:….
- Line 102, add a summary sentence indicating how do you collect samples of “Frass”.
- Line 231, please include the weight information of insects during the trial into the main body of the manuscript, it is key information proving that the LMW may tolerate aflatoxin contamination.
Author Response
Dear reviewer,
We thank you for your kind words and review of our manuscript. Below we have responded to your comments indicated in the manuscript.
Kind regards,
On behalf of my co-authors,
Nathan Meijer
“ Alphitobius diaperinus” or “Less Mealworm”, both names work but it needs to be consistent throughout the manuscript to improve readability, the name “ Alphitobius diaperinus” should be replaced with “lesser Mealworm”, and Lesser Mealworm as a name of species should be capitalized.
In the manuscript, we refer to the name of the species as Alphitobius diaperinus in a few instances. In principle, we agree with the reviewer that consistency would be improved if we were to refer to the species as Lesser Mealworm in all cases. However, the term Lesser Mealworm only refers to the larval stage of this species – and is therefore not applicable to pupae and beetles, which were also in scope. In those instances where we use Alphitobius diaperinus, we specifically mean to refer not only to the larval stage. Lesser Mealworm has been capitalized in all instances as requested.
Experiment procedures, add a summary sentence indicating how many replicates were used in this study at Line 88 before “All insects:….
The number of replicates was added as requested.
Line 102, add a summary sentence indicating how do you collect samples of “Frass”.
A description of how the frass was collected has been added.
Line 231, please include the weight information of insects during the trial into the main body of the manuscript, it is key information proving that the LMW may tolerate aflatoxin contamination.
We agree with the reviewer and have added the table containing weight information of insects to the manuscript. I now see that the table containing this information had initially mistakenly been omitted from the supplementary materials, so this data is now included.